# Transforming Growth Factor-β3/Chitosan Sponge (TGF-β3/CS) Facilitates Osteogenic Differentiation of Human Periodontal Ligament Stem Cells

**DOI:** 10.3390/ijms20204982

**Published:** 2019-10-09

**Authors:** Yangfan Li, Zhifen Qiao, Fenglin Yu, Huiting Hu, Yadong Huang, Qi Xiang, Qihao Zhang, Yan Yang, Yueping Zhao

**Affiliations:** 1Institute of Biomedicine and Guangdong Provincial Key Laboratory of Bioengineering Medicine, Jinan University, Guangzhou 510632, Chinatydhuang@jnu.edu.cn (Y.H.);; 2Department of Stomatology, Jinan University Medical College, Guangzhou 510632, China; ypzhao6026@gmail.com

**Keywords:** transforming growth factor-β3, chitosan sponge, human periodontal ligament cells, osteogenic differentiation

## Abstract

Periodontal disease is the main reason for tooth loss in adults. Tissue engineering and regenerative medicine are advanced technologies used to manage soft and hard tissue defects caused by periodontal disease. We developed a transforming growth factor-β3/chitosan sponge (TGF-β3/CS) to repair periodontal soft and hard tissue defects. We investigated the proliferation and osteogenic differentiation behaviors of primary human periodontal ligament stem cells (hPDLSCs) to determine the bioactivity and potential application of TGF-β3 in periodontal disease. We employed calcein-AM/propidium iodide (PI) double labeling or cell membranes (CM)-Dil labeling coupled with fluorescence microscopy to trace the survival and function of cells after implantation in vitro and in vivo. The mineralization of osteogenically differentiated hPDLSCs was confirmed by measuring alkaline phosphatase (ALP) activity and calcium content. The levels of COL I, ALP, TGF-βRI, TGF-βRII, and Pp38/t-p38 were assessed by western blotting to explore the mechanism of bone repair prompted by TGF-β3. When hPDLSCs were implanted with various concentrations of TGF-β3/CS (62.5–500 ng/mL), ALP activity was the highest in the TGF-β3 (250 ng/mL) group after 7 d (*p* < 0.05 vs. control). The calcium content in each group was increased significantly after 21 and 28 d (*p* < 0.001 vs. control). The optimal result was achieved by the TGF-β3 (500 ng/mL) group. These results showed that TGF-β3/CS promotes osteogenic differentiation of hPDLSCs, which may involve the p38 mitogen-activated protein kinase (MAPK) signaling pathway. TGF-β3/CS has the potential for application in the repair of incomplete alveolar bone defects.

## 1. Introduction

Periodontal disease is a chronic inflammatory condition that affects the supporting tissues around teeth, resulting in periodontal tissue breakdown or tooth loss in severe cases. Being highly prevalent among adults, periodontal disease is receiving increased attention from researchers and clinicians [1]. Therefore, repairing periodontal support tissues, such as alveolar bone, is indispensable for the treatment of periodontal disease. With the increasing popularity of dental implant surgery, the lack of bone mass in patients with periodontitis has limited the need for implant surgery, and the need to repair alveolar bone defects is increasing [2,3].

The current clinical techniques mainly used for the treatment of alveolar bone defects [4] are bone grafting and guided bone regeneration (GBR). Autologous bone grafting is considered to be the gold standard for bone repair [5,6,7,8], but it has many limitations such as long operation and recovery times, possible insufficient bone mass in patients, and numerous complications [9,10]. GBR uses a barrier to provide space that is filled with new bone to compensate for deficiencies [11,12]. However, GBR has shortcomings such as immune rejection and a poor morphological structure and mechanical properties [13,14,15]. In recent years, periodontal tissue engineering technology, which is characterized by the use of stem cells, bioactive molecules (e.g., growth factors), and scaffold materials as the three basic elements, has provided a new solution to reconstruct alveolar bone defects.

Transforming growth factor-β3 (TGF-β3) has been used for cartilage repair, tissue regeneration, and wound healing in vivo [16,17,18]. TGF-β3 facilitates matrix formation, immunity, and maintenance of stem cell characteristics [19]. TGF-β promotes the proliferation and early differentiation of mesenchymal stem cells (MSCs) into osteoblasts, chondrocytes, adipocytes, and tendon cells [20]. TGF-β3 also recruits endogenous MSCs to initiate bone regeneration [21,22,23,24]. TGF-β3 induces endochondral bone formation [25] and completes bone remodeling [26]. It might also play a profound role in some osteogenic stages. Therefore, TGF-β3 can be considered as a selective bioactive molecule to repair alveolar bone defects. Moioli et al. [27] demonstrated that autologous MSCs and controlled-release TGF-β3 reduce surgical trauma due to local osteotomy. However, TGF-β3 is easily degraded and inactivated. TGF-β3 diffuses easily from sites after topical periodontal tissue engineering application. Therefore, it is necessary to develop suitable carrier material. Chitosan is a natural polymer material that effectively promotes wound healing and early osteogenesis after tooth extraction [28]. We have previously developed a series of constructs with chitosan and its derivatives. Some applications of modified chitosan in the biomedical field have been reported [29]. Therefore, based on previous studies, chitosan may be a suitable carrier. Alveolar bone regeneration is enhanced by the addition of osteogenic cells to biomaterial scaffolds, which reduces treatment time and results in better outcomes and increased patient comfort [30]. Periodontal ligament stem cells (PDLSCs) promote osteoblastic and osteoclastic differentiation of osteoblasts and form an ectopic cementum/ligament-like complex [31]. Kim et al. [32] reported that the growth and induction of PDLSCs promote regeneration of cementum and the periodontal ligament to enable easy root fixation and resorption. Because PDLSC are suitable stem cells for periodontal tissue regeneration, we employed hPDLSCs to validate TGF-β3 in the regeneration of alveolar bone defects.

Considering the literature and results of our previous study, we hypothesized that TGF-β3 may promote osteogenic differentiation of hPDLSCs via the p38 mitogen-activated protein kinase (MAPK) pathway. PDLSCs undergo osteogenic differentiation through the MAPK signaling pathway [33,34,35]. The signal transduction mediated by TGF-β3 in osteogenic differentiation and bone regeneration [36] specifically occurs through both canonical Smad-dependent pathways (TGF-β ligands, receptors, and Smads) and noncanonical Smad-independent signaling pathways (e.g*.,* p38 MAPK pathway). Thus, we measured the levels of COL I, ALP, TGF-βRI, TGF-βRII, and Pp38/p38 by western blotting to validate these assumptions.

Worldwide, periodontitis affects the quality of life of the middle-aged population in terms of oral functioning. Unfortunately, no current clinical periodontal treatments can heal the defects in the affected region or regenerate lost periodontal tissue to a normal structure and functionality. It is obvious that there is a clinical need for such treatments and a vast patient demand [37]. The aim of this study was to examine the changes in proliferation and differentiation of hPDLSCs in TGF-β3/CS and explore the underlying mechanisms to repair defects in periodontal bone tissue.

## 2. Results

### 2.1. Preparation and Characterization of Transforming Growth Factor-β3/Chitosan Sponge (TGF-β3/CS)

As shown in Figure 1A, CS had a regular appearance and smooth surface. In the SEM image, TGF-β3/CS showed a three-dimensional (3D) porous network structure and interpenetrating pore structures resulting in a large internal surface area (Figure 1B). The pore size of the TGF-β3/CS was 156.95 ± 18.21 μm, the water absorption was 2347% ± 201%, the swelling ratio was 52.67% ± 12.42%, and the porosity was 85.65% ± 3.5%.

As shown in Figure 1C, TGF-β3 was stably released from CS at predetermined time points, cumulatively released from CS, and continued to act on cells (Figure 1D). The biocompatibility of TGF-β3/CS was evaluated by 3-(4,5-Dimethylthiazol-2-yl)-2,5-diphenyltetrazolium bromide (MTT) assays. Compared with the control group, the scaffold had no obvious cytotoxicity (*p* > 0.05; Figure 1E). Therefore, CS is a suitable carrier of TGF-β3, ensuring sustained and stable release of TGF-β3 in vivo and in vitro.

After hPDLSCs were cultured on TGF-β3/CS for 3 d, hPDLSCs grew well. The cell structure was intact, and there were about 95% viable cells (Figure 2A, green) and 5% dead cells (Figure 2A, red). TGF-β3/CS with hPDLSCs prestained with cell membranes (CM)-Dil was implanted subcutaneously into Sprague Dawley rats for 7, 14, and 21 d to observe the growth of cells (Figure 2B). Viable cells were observed after 21 d, indicating that hPDLSCs in TGF-β3/CS survived well in animals.

After hPDLSCs were seeded in TGF-β3/CS for 7 d, hPDLSCs displayed good adhesion and extension on the surface of CS, the cells adhered to each other to form a monolayer, and some cells crossed the pores of the porous sponge (Figure 2C). After inoculation of hPDLSCs on CS with various concentrations of TGF-β3, we observed that hPDLSCs grew and proliferated on CS (*p* < 0.001 vs. control; Figure 2D).

### 2.2. TGF-β3 Does not Affect Growth or Proliferation of hPDLSCs But Facilitates Their Osteogenic Differentiation

The relative proliferation rate of hPDLSCs cultured with various concentrations of TGF-β3 for 24, 48, and 72 h was examined by MTT assays. The relative proliferation rates depended on time, but they did not change with the concentration of TGF-β3. TGF-β3 did not significantly promote or inhibit the growth or proliferation of hPDLSCs within the concentration range (*p* > 0.05, vs. control), and the highest concentration showed no significant toxicity in hPDLSCs (Figure 3A). However, as shown in Figure 3B, TGF-β3 promoted alkaline phosphatase (ALP) secretion and calcium deposition of hPDLSCs. When the loading of TGF-β3 was 250 or 500 ng/mL, ALP expression and calcium in hPDLSCs were upregulated significantly. The optimal concentration range of TGF-β3 to promote cell differentiation was 250–500 ng/mL.

### 2.3. TGF-β3/CS Facilitates Osteogenic Differentiation of hPDLSCs

hPDLSCs were inoculated on CS with various concentrations of TGF-β3 (62.5, 250, and 500 ng/mL), and ALP values were determined for 3, 7, and 14 d after osteoblastic induction (Figure 4A). We observed no significant differences among TGF-β3/CS groups compared with the CS group at 3 d, but TGF-β3/CS groups exhibited significant differences after 7 and 14 d (*p* < 0.05, vs. control; Figure 4B).

Calcium content was measured after osteogenic induction for 14, 21, and 28 d. No significant differences we observed among the groups at 14 d, but all TGF-β3/CS groups had significantly higher levels of calcium content at 21 and 28 d (*p* < 0.05, vs. control). The optimal results were observed in the TGF-β3 (500 ng/mL)/CS group (*p* < 0.001, vs. control) (Figure 4C).

### 2.4. TGF-β3 Promotes Osteogenic Differentiation of hPDLSCs via the p38 MAPK Pathway

To verify the effect of TGF-β3 on osteogenic differentiation of hPDLSCs, we examined changes in osteogenesis-associated proteins (COL I, COL II, and ALP) and pathway proteins (TGF-βRI, TGF-βRII, p38, Pp38, and Runx2) over the induction time and the TGF-β3 level by western blotting (Figure 5A and Figure 6B). There was a statistically significant difference in the increase of COL I expression in the TGF-β3 (500 ng/mL) induction group after 7 and 14 d of induction (*p* < 0.001, vs. control), whereas the other induction groups showed no statistical differences (*p* > 0.05, vs. control). The level of COL I expression induced after 14 d was significantly higher than that induced after 7 d (Figure 5B).

COL II expression was significantly increased in TGF-β3 (125, 250, and 500 ng/mL) induction groups after 7 d of induction (*p* < 0.001, vs. control). After 14 d, there was no statistical difference in the induction groups (*p* > 0.05, vs. control). Compared with the results at day 7, COL II expression in TGF-β3 (125, 250, and 500 ng/mL) induction groups at day 14 was reduced significantly (Figure 5C).

ALP expression was significantly decreased in TGF-β3 (500 ng/mL) induction groups after 7 and 14 d of induction (*p* < 0.01, vs. control), and no significant difference was found among the other groups (*p* > 0.05, vs. control). The ALP expression induced at 14 d was significantly lower than that at 7 d (Figure 5D).

These results showed that as the induction time increased, COL I in the TGF-β3 (500 ng/mL) induction group increased significantly, and COL II and ALP decreased significantly.

TGF-βRI expression was significantly increased in TGF-β3 (500 ng/mL) induction groups after 7 and 14 d of induction (*p* < 0.001, vs. control). The expression of TGF-βRI induced for 14 d was significantly higher than that induced for 7 d (Figure 6C). TGF-βRII expression in the TGF-β3 (500 ng/mL) induction group was significantly increased after 7 and 14 d of induction (*p* < 0.001, vs. control). The TGF-βRII expression induced for 14 d was significantly lower than that induced for 7 d (Figure 6D). No significant difference was found in the expression of t-p38 in the induction groups after 7 and 14 d of induction (*p* > 0.05, vs. control) (Figure 6E), whereas Pp38 expression was significantly increased in the TGF-β3 (500 ng/mL) induction group (*p* < 0.001, vs. control). Pp38 induced for 14 d was significantly higher than that induced for 7 d (Figure 6F). Runx2 expression in the TGF-β3 (500 ng/mL) induction group was significantly increased after 7 d of induction (*p* < 0.001, vs. control). After 14 d, no statistical difference was found among the induction groups (*p* > 0.05, vs. control). Runx2 expression induced for 14 d was significantly lower than that induced for 7 d (Figure 6G).

These results showed that as the induction time increased, TGF-βRI and Pp38 in the TGF-β3 (500 ng/mL) induction group increased significantly, TGF-βRII and Runx2 decreased significantly, and t-p38 was unchanged (Figure 6C). Therefore, Pp38/t-p38 showed an upregulated trend, indicating that TGF-β3 significantly increased osteogenic differentiation of hPDLSCs.

## 3. Discussion

The aim of periodontal tissue engineering is to regenerate the supporting tissue of teeth through a combination of proper biomaterials, such as growth factors and scaffolds, which stimulate cells and signaling molecules to produce new, healthy tissue.

PDLSCs have broad application prospects as odontogenic seed cells in periodontal tissue engineering and regenerative medicine because of their biosafety and odontogenic properties. Biomaterials that repair alveolar bone defects require stable biological properties and good biocompatibility. Many studies had examined the use of TGF-β3 for cartilage repair, tissue regeneration, and wound healing in vivo [38].

In this study, we prepared a TGF-β3/CS freeze-dried sponge, a high-porosity network material with good water absorption and swelling rates. It mimics the natural extracellular microenvironment of dental tissue and promotes the adhesion, proliferation, and differentiation of hPDLSCs.

Morphology and calcein-AM/ propidium iodide (PI) double-staining results showed that hPDLSCs spread into a typical fusiform shape on TGF-β3/CS and continued to grow and proliferate. The cytotoxicity of TGF-β3/CS was evaluated using an extraction test, showing that TGF-β3/CS had no obvious cytotoxicity. In vitro experiments revealed that TGF-β3/CS did not significantly promote or inhibit the growth and proliferation of hPDLSCs, but it promoted osteogenic differentiation of hPDLSCs. When hPDLSCs prestained with CM-Dil were loaded on TGF-β3/CS and implanted subcutaneously into Sprague Dawley rats, hPDLSCs were detected in rats during the course of experiments. This result indicated that TGF-β3/CS had stable biological properties and good biocompatibility, suggesting its application as a biomedical material in tissue engineering. Osteogenic differentiation of cells is a complex process involving cell proliferation, extracellular matrix maturation, and mineralization [39]. ALP is an enzyme on the cell membrane, which catalyzes the hydrolysis of phosphate esters, and is one of the major indicators to evaluate the degree of early osteogenic differentiation of cells. ALP activity in the TGF-β3 (250 ng/mL)/CS induction group on day seven was significantly higher than that in the other groups. Compared with day 7, ALP activity increased at day 14, which was not consistent with the results of ALP expression measured by western blotting. This may be due to ALP being a marker of early osteogenic differentiation of cells, and when cells begin to enter the late stage of osteogenic differentiation, the ALP secreted by cells may accumulate in the culture supernatant, resulting in an increase in ALP content. Alizarin red staining is a common method to identify advanced osteogenic differentiation of cells [33]. hPDLSCs were cultivated in osteogenic differentiation medium and stained with alizarin red after 14 days, showing that TGF-β3 promoted osteogenic differentiation. However, hPDLSCs on TGF-β3/CS could not be stained with alizarin red to detect the calcium ion content because the high-porosity chitosan material had an adsorption effect on the alizarin red solution. Thus, the interference of the material itself could not be circumvented. Therefore, we detected and quantified osteocalcin of hPDLSC in the material using calcium colorimetry, indicating that TGF-β3/CS increased the level of calcium in cells. The TGF-β3 (500 ng/mL)/CS group had the highest calcium content, demonstrating that a high concentration of TGF-β3 is more favorable for osteogenic differentiation of hPDLSCs in the late stage of differentiation.

The methods to repair bone defects can be divided into intramembranous osteogenesis and endochondral ossification. The former induces osteoblasts to secrete a bone-like mineralized matrix, whereas the latter induces cartilage cells to produce cartilage matrix that gradually mineralizes to form a bone-like matrix. COL I plays an indispensable role in extracellular matrix maturation and the formation of mineralized nodules, and it is one of the markers of early osteogenic differentiation of cells [40]. COL II is a major component of hyaline cartilage and plays a key role in maintaining chondrocyte functions. It is also an important indicator of chondrogenic differentiation. TGF-β3 contributes to significant improvement of the formation of type II collagen, inducing and promoting cartilage differentiation [41]. According to the western blot results, COL I expression in all groups was upregulated with the prolongation of culture time, and the expression level of COL I in the TGF-β3 (500 ng/mL)-induced group was higher than that in the control group. COL II expression began to increase gradually, and expression of COL II at day seven in the TGF-β3-induced group was significantly higher than that in the COL I group. This may be due to TGF-β3 promoting cell differentiation into cartilage in the early stage of cell differentiation. However, the expression levels of COL II were downregulated in all induction groups, while the expression levels of COL I were upregulated. This may be due to the initiation of endochondral ossification in the TGF-β3-induced group and entry into the mineralization phase of the cells. Endochondral ossification [42] is the process of depositing collagen and noncollagen on cartilage for mineralization. The cartilage expands, matures, and hypertrophies, and then the hypertrophic cartilage matrix is gradually replaced by trabecular bone. A decrease in the expression level of COL II has been reported to indicate a stage in which endochondral ossification enters cartilage matrix calcification [43]. Another study [44] found that TGF-β3 induces ossification of human adipose stromal cells into bone tissue in a rat bone defect model, and the trend in COL I and COL II protein expression was essentially consistent with the results of this study.

To confirm the validity of our hypothesis, the levels of TGF-βRI, TGF-βRII, and Pp38/t-p38 were measured by western blotting. Following TGF-β3 induction, p38 MAPK pathways converged at Runx2 to control hPDLSC differentiation. However, Runx2 is the earliest and continuously expressed protein in the process of cell osteogenesis, marking the beginning of osteogenic differentiation. As shown by western blotting, Runx2 expression at day 14 was significantly lower than that at day 7. This trend was consistent with the results reported by Paolella et al. [45]. Another study [46] has shown that Runx2 plays an important role in coordinating the multiple signals involved in osteoblast differentiation and is a specific transcriptional regulator necessary for osteoblast differentiation and bone formation. An active p38 MAPK pathway was verified by western blotting, but the associated protein TAK1 was not detected, potentially because other pathways can bypass this protein, enter downstream, and finally result in osteogenesis, but this remains to be verified.

In summary, TGF-β3 does not affect the normal proliferation of hPDLSCs (*p* > 0.05), but within a certain concentration range (62.5–500 ng/mL), 500 ng/mL TGF-β3 promotes the osteogenic differentiation of hPDLSCs via activation of the p38 MAPK pathway (*p* < 0.05). The prepared TGF-β3/CS has a good water absorption rate (2347% ± 201%), swelling ratio (52.67% ± 12.42%), and porosity (85.65% ± 3.5%), which are favorable for the adhesion, spreading, and growth of seed cells. It also has good biosafety and is an ideal carrier. TGF-β3/CS promotes the osteogenic differentiation of hPDLSCs, and the combination of the two is expected to be used for the repair of alveolar bone defects.

## 4. Materials and Methods

### 4.1. Preparation and Characterization of TGF-β3/CS

CS (molecular weight, 3.6 ×10^3^ Da; deacetylation degree, 50%) was purchased from Zhengzhou Kerui Fine Chemical Co., Ltd. (Zhengzhou, China). TGF-β3 was supplied by the Biopharmaceutical R&D Center of Jinan University (Guangzhou, China). All other chemical reagents obtained from Shanghai Lingfeng were of analytical grade. Briefly, after CS (2% *w*/*v*) was lyophilized, it was placed in a 95% ethanol solution for 2 h, and then the ethanol was discarded. Next, the CS was immersed in a 10% sodium hydroxide solution for 2 h and then repeatedly cleaned with deionized water until the pH was about 7. The lyophilized sponge-like biomaterial was sterilized and stored for later use. TGF-β3 at various concentrations (0, 62.5, 250, and 500 ng/mL) was loaded on CS dropwise.

The water absorption rate, swelling ratio, and porosity of CS were calculated as reported previously [47]. SEM observation of CS was performed as reported previously [48]. Briefly, the prepared CS was adhered to the copper stage with a conductive paste. Before the observation, the sample was subjected to gold sputtering using a gold spray carbonator, and the micromorphology of CS was observed by scanning electron microscopy (SEM, XL30; Philips, Amsterdam, The Netherlands).

#### Release Profile of TGF-β3 from CS

The release of TGF-β3 from CS was measured by an ELISA (CUSABIO, Wuhan, China). The scaffold (three replicates/group) was placed in a 1.5 mL Eppendorf tube, and then 1 mL minima essential medium (MEM) was added, followed by incubation at 37 °C for 35 d. Then, 1 mL of MEM was collected, and 1 mL fresh MEM was added at days 1, 3, 7, 14, 21, 28, and 35. The samples were stored at –80 °C until ELISA measurements. The ELISA was performed according to the manufacturer’s protocol. Light absorbance was read with a microreader (Thermo Lab systems, Waltham, MA, USA) at a wavelength of 450 nm.

### 4.2. Culture of hPDLSCs

hPDLSCs were obtained from tissues attached to the middle third of the tooth root from healthy patients 15–20 years old (five men and five women), as described previously [49], who were undergoing orthodontic treatment at the First Affiliated Hospital of Jinan University. All experimental protocols were approved by the Ethics Committee of Jinan University (Guangdong, China) (Approval number: 2019228, 28 February 2019). The tissues were minced, digested, and cultured in α-MEM (Gibco, New York, NY, USA), supplemented with 10% fetal bovine serum (FBS, Gibco), 100 mg/mL streptomycin, and 100 U/mL penicillin (MDBio, Shanghai, China) at 37 °C with 5% CO_2_. The medium was changed every 3 d, and hPDLSCs at passages (P) 3–5 were used in the following experiments.

### 4.3. Bioactivity and Biocompatibility Assays in Vitro

#### 4.3.1. Biocompatibility Analysis of TGF-β3/CS Extract

The cytotoxicity of TGF-β3/CS was evaluated by an extraction test. The ratio between the sample surface and volume of the medium was 0.5 cm^2^/mL. In brief, hPDLSCs were cultured in a 96-well plate at a density of 1 × 10^4^ cells/well in MEM and 10% FBS for 24 h. The cells were then divided into five groups and treated with 25%, 50%, 75%, or 100% CS extract, including a blank control of the extract alone. At days 1, 2, and 3 of incubation, the proliferative capacity of the cells in each group was examined by MTT assays [47].

#### 4.3.2. Cell Proliferation Assay of hPDLSCs Loaded on TGF-β3/CS in Vitro

Five concentrations of TGF-β3 (0, 12, 62.5, 250, and 1000 ng/mL) were loaded on CS. hPDLSCs were seeded in 24-well plates containing TGF-β3/CS at a density of 2 × 10^4^ cells/well. The viability of the cells was determined by CCK-8 assays (Enhanced Cell Counting Kit-8, Beyotime, Shanghai, China).

hPDLSCs were seeded in 96-well plates at a density of 5 × 10^3^ cells/well and then treated with various concentrations of TGF-β3 (0, 12, 62.5, 250, 500, and 1000 ng/mL). Cell proliferation was evaluated by MTT assays.

A microplate reader (Thermo Lab systems, Waltham, MA, USA) was used to detect the absorbance at 570 nm (MTT assay) or 450 nm (CCK-8 assay) after shaking samples for 5 min. Each assay was performed in triplicate.

#### 4.3.3. Growth of hPDLSCs Implanted in TGF-β3/CS in Vitro

To further study cell growth on TGF-β3/CS, hPDLSCs were cultured on TGF-β3/CS for 3 d and then stained with calcein AM/PI (Calcein-AM/PI Double Stain Kit, Shanghai, China), followed by fluorescence microscopy (LSM700, Zeiss, Jena, Germany) to observe the staining by detecting red (535 nm, AM) and green (490 nm, PI) fluorescence.

### 4.4. Growth of hPDLSCs Implanted in TGF-β3/CS in Vivo

hPDLSCs prestained with CM-Dil (BestBio, Shanghai, China) were loaded on TGF-β3/CS and then implanted subcutaneously into Sprague Dawley rats (280 ± 20 g, male, *n* = 15; no. 37009200016139). Animals were sacrificed at 7, 14, and 21 d to observe the cells by detecting red (553 nm, CM) fluorescence using the LSM700 fluorescence microscope. All procedures for animal handling were based on the principles (ref 006939801/2010-00810) of Laboratory Animal Care formulated by the National Society for Medical Research and approved by the Animal Care and Experiment Committee of University of Jinan University, Guangzhou, China. Experiments complied with the National 132 Institutes of Health guide for the care and use of laboratory animals (NIH Publication No. 8023, revised 1996). Samples were harvested with the surrounding tissue followed by examination under the LSM700 fluorescence microscope to observe the staining.

### 4.5. Induction of Osteogenic Differentiation in Vitro

hPDLSCs (P3) were seeded in 12-well plates at 5 × 10^4^ cells/well and cultured until 80% confluence. Then, hPDLSCs were cultivated in osteogenic differentiation medium consisting of α-MEM containing 10^−8^ M dexamethasone (Sigma-Aldrich, St. Louis, MO, USA), 10 mM β-glycerophosphate (Sigma-Aldrich), 50 ng/mL ascorbic acid (Sigma-Aldrich), 10% FBS, and 1% penicillin–streptomycin. The osteogenic medium was changed every 2 d.

Induction of hPDLSCs was completed with various concentrations of TGF-β3 (0, 62.5, 250, and 500 ng/mL). After cultivation for 3 and 7 d, hPDLSCs were stained with an alkaline phosphatase (ALP) staining kit (Beyotime Institute of Biotechnology, Shanghai, China) to test their early osteogenic differentiation capacity. After cultivation for 14 d, the hPDLSCs were stained with alizarin red (Cyagen Biosciences, CA, USA) to test their late osteogenic differentiation capacity.

Four concentrations of TGF-β3 (0, 62.5, 250, and 500 ng/mL) were loaded in CS. After cultivation for 3, 7, and 14 d, the ALP activity of hPDLSCs on TGF-β3/CS was determined using the ALP staining kit. To detect and quantify osteocalcin in hPDLSCs at the late stage of each group of materials, a calcium colorimetric assay (Sigma-Aldrich) was applied after cultivation for 14, 21, and 28 d.

### 4.6. Western Blot Analysis

After 7 and 14 d of cultivation, cells were digested and collected. The cell pellets were lysed in radio immunoprecipitation assay (RIPA) buffer (Cell Signaling Technology, Beverly, MA, USA) on ice for 30 min and then centrifuged at 12,000 rpm for 30 min at 4 °C. The supernatants were collected, and the protein concentrations were measured with a bicinchoninic acid (BCA) protein assay kit (Life Technologies, Carlsbad, CA, USA), according to the manufacturer′s instructions. Sodium dodecyl sulfate polyacrylamide gel electrophoresis and immunoblotting were conducted according to standard protocols and visualized using the ChemiDoc-It Imaging System (UVP, Upland, MA, USA). Antibodies against TGF-βRI, TGF-βRII, p38, Pp38, COLI (Affinity Biosciences, Cincinnati, OH, USA), ALP (alkaline phosphatase), COLII (Abcam, Cambridge, MA, USA), GAPDH, and an HRP-conjugated secondary antibody (Cell Signaling Technology, Boston, MA, USA) were used.

### 4.7. Statistical Analysis

All data are expressed as the mean ± standard deviation (SD) of at least three independent experiments. Statistical analyses were performed using GraphPad Prism 6 software (GraphPad Software Inc., La Jolla, CA, USA). Differences among more than two groups were analyzed by one-way ANOVA, followed by the Tukey HSD comparison test. Values of *p* < 0.05 were considered as statistically significant.

## 5. Conclusions

TGF-β3 has no negative effect on the proliferation of hPDLSCs, and an appropriate concentration of TGF-β3 promotes osteogenic differentiation of hPDLSCs via activation of the p38 MAPK pathway. The prepared TGF-β3/CS has a good water absorption rate, swelling ratio, and porosity. It is favorable for the adhesion, spreading, and growth of seed cells; it has good biosafety; and it conforms to the medical standard of biological materials. In addition, the optimal concentration of TGF-β3 used on CS is 500 ng/mL. TGF-β3/CS promotes osteogenic differentiation of hPDLSCs, and the combination of the two is expected to be used for the repair of alveolar bone defects.

## Figures and Tables

**Figure 1 ijms-20-04982-f001:**
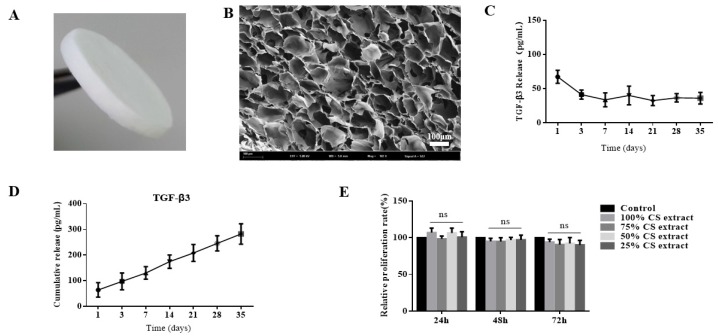
Characterization of transforming growth factor-β3/chitosan sponge (TGF-β3/CS) and release of TGF-β3 from CS. (**A**). Photograph of CS. (**B**). Scanning electron microscopy (SEM) image of CS (×10^2^). Scale bar represents 100 μm. (**C**). Release curve of TGF-β3 from CS (mean ± SD; *n* = 3). (**D**). Cumulative release of TGF-β3 from CS (mean ± SD; *n* = 3). (**E**). Cytotoxicity of CS measured by 3-(4,5-Dimethylthiazol-2-yl)-2,5-diphenyltetrazolium bromide (MTT) assays (mean ± SD; *n* = 5). Blank control group: cells were cultured with only the medium; CS extract group: cells were cultured with 25%, 50%, 75%, or 100% CS extract (ns means no significant differences, *p* >0.05 vs. control).

**Figure 2 ijms-20-04982-f002:**
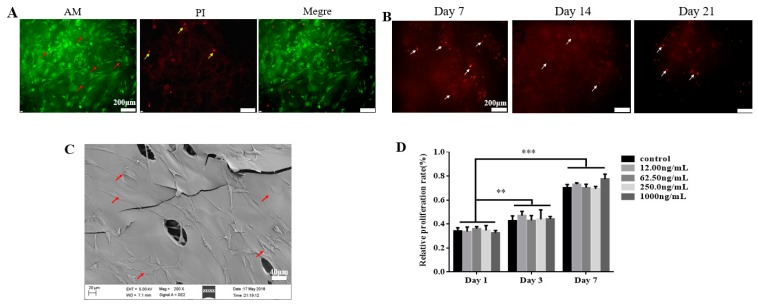
Effect of TGF-β3/CS on the growth and proliferation of primary human periodontal ligament stem cells (hPDLSCs). (**A**) Calcein-AM/ propidium iodide (PI) double staining of hPDLSCs on TGF-β3/CS after 3 d of culture in vitro. Live cells (red arrow) are stained by AM (green), and dead cells (yellow arrow) are stained by PI (red) (×100). Scale bar represents 100 μm. (**B**) TGF-β3/CS with hPDLSCs (white arrow) implanted in Sprague Dawley rats for 7, 14, and 21 d and then stained with CM-Dil (red). Cell survival was observed under a fluorescence microscope (×100). Scale bar represents 100 μm. (**C**) SEM photomicrographs of hPDLSCs (red arrow) in CS for 7 d (×200). Scale bar represents 40 μm. (**D**) hPDLSC growth in TGF-β3/CS was measured by CCK-8 assays (mean ± SD; *n* = 5). ** *p* < 0.01 vs. control; *** *p* < 0.001 vs. control.

**Figure 3 ijms-20-04982-f003:**
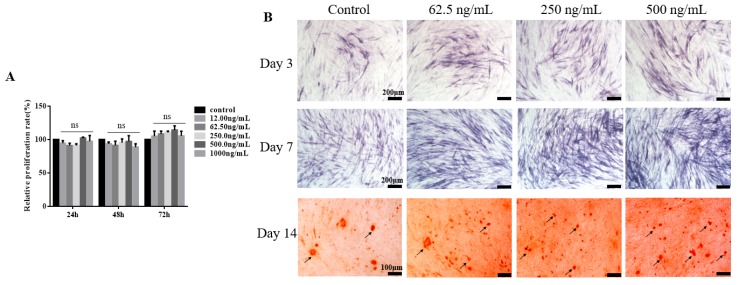
TGF-β3 does not affect the growth or proliferation of hPDLSCs, but it promotes their osteogenic differentiation. (**A**). Relative proliferation rates of hPDLSCs cultured with various concentrations of TGF-β3 for 24, 48, and 72 h determined by MTT assays (*n* = 5). (**B**). Alkaline phosphatase (ALP) staining (purple) was used to detect the ALP activity of hPDLSCs cultured with various concentrations of TGF-β3 after 3 and 7 d of osteogenic induction. Alizarin red staining (red) was used to detect the calcium content of hPDLSCs cultured with various concentrations of TGF-β3 after 14 d of osteogenic induction. Black arrow shows calcium deposition. ns means no significant differences, *p* >0.05 vs. control.

**Figure 4 ijms-20-04982-f004:**
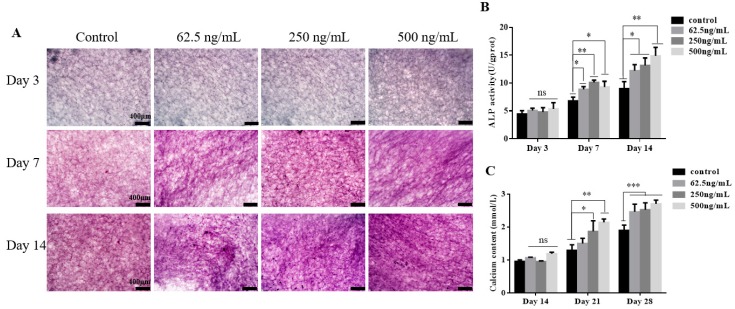
Effect of TGF-β3/CS on osteogenic differentiation of hPDLSCs. (**A**). ALP staining (purple) was used to detect the ALP activity of hPDLSCs on CS with various concentrations of TGF-β3 after 3, 7, and 14 d of osteogenic induction. (**B**). Detection of ALP activity in hPDLSCs on CS with various concentrations of TGF-β3 after 3, 7, and 14 d of osteogenic induction. (**C**). Determination of calcium in hPDLSCs on CS with various concentrations of TGF-β3 after 14, 21, and 28 d of osteogenic induction (*n* = 3). ns means no significant differences, *p* >0.05 vs. control; * *p* < 0.05 vs. control; ** *p* < 0.01 vs. control; *** *p* < 0.001 vs. control.

**Figure 5 ijms-20-04982-f005:**
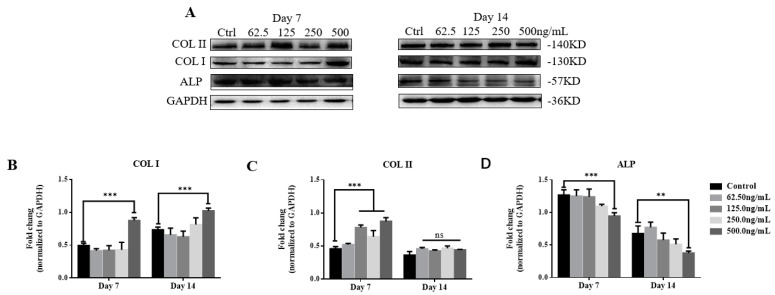
Expression and analysis of proteins associated with osteogenic differentiation. (**A**–**D**) After osteogenic induction of hPDLSCs with various concentrations of TGF-β3 for 7 and 14 d, expression of osteogenic proteins (COL I, COL II, and ALP) was detected by (**A**) western blotting and analyzed by grayscale scanning for (**B**) COL I, (**C**) COL II, and (**D**) ALP (*n* = 3). ns means no significant differences, *p* >0.05 vs. control; ** *p* < 0.01 vs. control; ****p* < 0.001 vs. control. Abbreviations: Col I, type I collagen; Col II, type II collagen; ALP, alkaline phosphatase.

**Figure 6 ijms-20-04982-f006:**
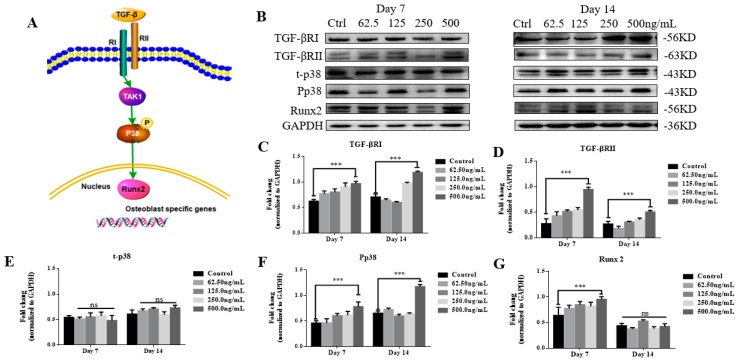
Mechanism of TGF-β3 in osteogenic differentiation of hPDLSCs based on this study. (**A**) Schematic representation of the mechanism of TGF-β3 in osteogenic differentiation of hPDLSCs. (**B**) After osteogenic induction of hPDLSCs with various concentrations of TGF-β3 for 7 and 14 d, expression of osteogenic pathway proteins was detected by (B) western blotting and analyzed by grayscale scanning for (**C**) TGF-βRI, (**D**) TGF-βRII, (**E**) t-p38, (**F**) Pp38, and (**G**) Runx2 (*n* = 3). ns means no significant differences, *p* >0.05 vs. control; *** *p* < 0.001 vs. control. Abbreviations: TGF-βRI, transforming growth factor-β receptor I; TGF-βRII, transforming growth factor-β receptor II; t-p38, total p38; Pp38, phosphorylated p38; Runx2, runt-related transcription factor 2.

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
