# Peer review of "Transforming Growth Factor-β3/Chitosan Sponge (TGF-β3/CS) Facilitates Osteogenic Differentiation of Human Periodontal Ligament Stem Cells"

_ijms, 2019, doi:10.3390/ijms20204982_

Round 1

Reviewer 1 Report

Dear Authors, 

congratulations for Your study. However I have some suggestions before the manuscript can be suitable for publication. 

Introduction: 

I would explain a bit more on why the periodontal disease can lead to the tooth loss (due for example to the negative effects of the inflammatory diseases 

this paper might help You http://apps.webofknowledge.com/InboundService.do?product=WOS&Func=Frame&DestFail=http%3A%2F%2Fwww.webofknowledge.com%3FDestParams%3DUT%253DWOS%25253A000464697500040%2526customersID%253DCINECA_CRIS%2526smartRedirect%253Dyes%2526action%253Dretrieve%2526mode%253DFullRecord%2526product%253DCEL%26SrcAuth%3DCINECA_CRIS%26SrcApp%3DCINECA_CRIS%26DestApp%3DCEL%26e%3DVX4hjF%252BMZwGzs11jccbfY1F5ITh%252FQv5zfbfFgfiAYprui8TKE8Tr5LTDBbCmBpaV&SrcApp=CINECA_CRIS&SrcAuth=CINECA_CRIS&SID=D339sRLJkuCpDETjhHH&customersID=CINECA_CRIS&smartRedirect=yes&mode=FullRecord&IsProductCode=Yes&Init=Yes&action=retrieve&UT=WOS%3A000464697500040

I miss the method of preparation of the CS for the SEM observation

I do not understand why You used a different type of cells for the MTT test

Results 

I miss the information of SEM pictures (magnification, distance etc)

Same comments on fluorescence I miss some info like wavelenght etc

How did You calculate the MTT data? I mean how did You arrived from the raw data to the percentage? Explain in the methods

I will add some info on the properties of the chitosan and the importance of the appropriate carrier 

This paper might help You

https://www.ncbi.nlm.nih.gov/pmc/articles/PMC6267118/

Author Response

Dear Editors and Reviewers,

Thank you for the letter and the reviewers’ comments concerning our manuscript entitled “Transforming Growth Factor-β3 Chitosan Sponge (TGF-β3/CS) Facilitates Osteogenic Differentiation of Human Periodontal Ligament Stem Cells” (ID: ijms-599436). Thanks for every editors/reviewers’ valuable comments and suggestions. Those comments are all valuable and very helpful for revising and improving our paper, as well as the important guiding significance to our researches. We have studied all the comments carefully and responded to them accordingly. We also have revised the manuscript based on all comments and suggestions. Revised portion are marked using the "Track Changes" function in Microsoft Word in the paper. We accepted all criticisms of reviewers on our spelling errors and grammar. We made significant changes to our manuscript, which rewrote many of the content. We also edited our manuscript by MDPI. We hope all the corrections will answer the questions and meet the requirements. We sincerely hope that editors/reviewers are willing to accept our explanation and our manuscript can be published in your excellent journal.

Thank you and best regards!

Yours sincerely,

Zhifen Qiao

Reviewer 2 Report

The authors represent a study on the use TGF-b chitosan sponge in the osteogenic differentiation of human periodontal ligament cells. The authors represent an innovative approach for periodontal regeneration, however, the fall very short from presenting their work in an attractive style. Below, the reviewer lists some comments regarding this manuscript:

1) Manuscript structure should be divided into (in vitro) and (in vivo). The authors do not present this clearly in the manuscript which creates alot of confusion to the reader. 

2) Instead of presenting the aim and objectives of the study at the end of the introduction, the authors are listing the conclusions of their study in the introduction. 

3) Conclusions are not clear: what is the best concentration of TGF-b to be used on this scaffold? 

4) Please specifiy the aims and objectives of your study in the introduction. In the discussion, the authors should structure their findings in correspondence to the aims (e.g. biocompatibility, cytotoxcitiy, chitosan perfomrance)

5) The authors do not discuss in depth about Chitosan properties whether in the introduction or discussion. As known, Chitosan does not have very strong mechanical properties. Why mechanical strength was not tested for example? How do we overcome this challenge when aiming at regenerating alveolar bone? (i.e. load-bearing area)

6) Why did you choose Chitosan specifically? Why not other biomaterial? Rationale for the use is not clear. 

7) The authors discuss (living) and (dead) cells just by mentioning. The authors need to describe this in numbers (i.e. percentage) 

8) In vitro experiments are usually repeated 3 times. The authors do mention if they repeated the experiments or not. 

9) The in vivo part of this study is under-described. Furthermore, why did the authors not make use of the in vivo model for meaningful histological analysis? This reviewer thinks that the in vivo model has been under-used in this study. 

10) What is the figure that appears after the abstract and before the introduction? No figure number and no figure legend is mentioned.

11) English language needs to be revised throughout the whole manuscript. I will give few examples; e.g. in the introduction line 64, the sentence should be: TGF-B easily degrades. Line 65, it should be (is) instead of (was). Line 66, the sentence should be: and early osteogenesis after tooth extraction, which makes it an ideal carrier for TGF-B. 

12) In the discussion, line 229: long time; I would not consider 4 weeks a long time in rats (in vivo studies in rats consider 6 weeks or 9 weeks a long healing time)

13) Brief summary of your findings should be presented at the end of the discussion, with your statistically significant results.  

14) In the materials and methods, section 4.3.3: Animals were sacrificed at 2 and 4 weeks. Since you were mentioning the scaffold implantation in days (7, 14, 21 days), the sacrificing time points should be mentioned in days too. 

Author Response

(The authors gave the same response as above.)

Round 2

Reviewer 2 Report

The manuscript has very much improved from its previous version. The authors have addressed all the comments successfully.

Although the authors have sent their manuscript to MDPI for language check, there are still many linguistic errors, and thus, the manuscript should be checked again for English language before being published. The reviewer suggests that the authors should contact the language editor who revised their manuscript. I will give few examples:

1) In the introduction, page 3: "To validate the assumes" there is no sentence as such. I would use "to validate findings" or "to validate assumptions"

2) In the introduction, page 3: "and satisfied the needs of clinicians" should be "and satisfy the needs of clinicians"

3) In discussion page 9: "To confirms our hypothesis" should be "To confirm" 

Author Response

Dear Editors and Reviewers,

Thank you for the letter and the reviewers’ comments concerning our manuscript entitled “Transforming Growth Factor-β3 Chitosan Sponge (TGF-β3/CS) Facilitates Osteogenic Differentiation of Human Periodontal Ligament Stem Cells” (ID: ijms-599436). Thanks for every editors/reviewers’ valuable comments and suggestions. According to your comments, we will submit the manuscript to the Royal Society of Chemistry (https://rsc-editing.org/) for linguistic modification by native English speakers. However, the linguistic modification will be finished in seven days. So, we can’t return the manuscript in time. Would you like to give us more time? We hope to do our best to modify the manuscript until it can meet the requirements of yours. We sincerely hope our manuscript can be published in your excellent journal.

Thank you and best regards!

Zhifen Qiao